# Aggressive, Submissive, and Affiliative Behavior in Sanctuary Chimpanzees (Pan Troglodytes) During Social Integration

**DOI:** 10.3390/ani12182421

**Published:** 2022-09-14

**Authors:** Amy Fultz, Akie Yanagi, Sarah Breaux, Leilani Beaupre

**Affiliations:** 1Chimp Haven, 13600 Chimpanzee Place, Keithville, LA 71047, USA; 2Office of Academic Affairs, Niagara County Community College, 3111 Saunders Settlement Rd, Sanborn, NY 14132, USA; 3Department of Veterinary Resources, University of Louisiana at Lafayette-New Iberia Research Center, New Iberia, LA 70562, USA; 4Independent Researcher, Tumwater, WA 98512, USA

**Keywords:** chimpanzee, introduction, social integration, socialization, behavior

## Abstract

**Simple Summary:**

Chimpanzees moving into sanctuary settings from captive research settings have often lived in smaller groups at their previous facilities. However, chimpanzees in the wild live in large multi-male, multi-female groups that provide them with a greater variety of social opportunities. Chimp Haven introduces chimpanzees to each other to form larger, more complex groups. This has resulted in hundreds of introductions or social integrations over a ten-year period. There are very limited published data on social integrations in chimpanzees. The purpose of the current study is to determine how group size, location, rearing, age, and sex differences affect the chimpanzees’ affiliative, aggressive, and submissive behaviors during these social integrations to help inform future efforts at socializing chimpanzees in captive settings. The findings of the study are likely to assist us in determining which factors might be most important to consider for chimpanzee introductions.

**Abstract:**

Chimp Haven is a sanctuary for chimpanzees being retired from biomedical research and from facilities that can no longer care for them. Chimpanzees often live in smaller groups in captive settings; however, Chimp Haven integrates them into larger, more species-typical groups. Social integrations, the process of introducing unfamiliar chimpanzees to one another, are often complex in terms of logistics and can be stressful due to the territorial nature of the animals, reduced space in captivity, and the fact that these situations are engineered by humans. From 2005 to 2015, Chimp Haven conducted 225 social integrations including 282 chimpanzees (male: n = 135; female: n = 147). Each integration involved 2 to 26 chimpanzees (mean = 9) and their age ranged from < one year old to 59 years old (mean = 30). We collected data ad libitum during the first 60 min after doors were opened between unfamiliar chimpanzees. The chimpanzees’ affiliative, aggressive, and submissive behaviors were examined, comparing the subject’s sex, rearing history, location/enclosure type, and group size impacts on these behaviors. The subject’s sex, location, and group size were associated with the frequency of affiliative behaviors observed during social integration. All variables except for group size were associated with the frequency of aggressive behavior. The frequency of submissive behavior differed based on the subject’s sex, rearing history, and group size. We were unable to make comparisons between successful and unsuccessful integrations, as most of these integrations were successful.

## 1. Introduction

Over the last 17 years, chimpanzees have been moving into sanctuaries in unprecedented numbers [1,2]. Many of these chimpanzees have come from biomedical research facilities or situations where they were reared by or had close relationships with humans. In captivity, particularly in the past, chimpanzees have typically lived by themselves, in pairs or in smaller groups of up to five individuals with some exceptions. In a recent survey of research facilities, 63% of the chimpanzees lived in pairs or small groups; the mean group size was 4.6 chimpanzees, but group sizes ranged from 1 to 14 chimpanzees [3].

Chimpanzees are a very social species in the wild, living in communities of 20–150 individuals [4,5]. Chimpanzee communities do not typically spend all their time together but have a mammalian social dynamic structure typically referred to as fission–fusion where smaller groups often spend their days together and then larger groups come together at large feeding resources, for breeding, and to nest in the trees at night [6]. Although it would be difficult to create space in captivity for very large groups of hundreds of chimpanzees, in recent years, both the Association of Zoos and Aquariums (AZA) and the National Institutes of Health Working Group have recommended that captive facilities integrate their chimpanzees into larger groups to provide more complexity and social opportunities for the chimpanzees in the United States. Both suggest optimal group sizes of 7–9 chimpanzees [7,8]. The AZA Chimpanzee SSP (Species Survival Plan) recommends the maintenance of mixed sex social groups that contain multiple males and age diversity. With appropriate staff and facilities, the AZA also recommends that the size of a social group should meet or exceed three adult males and five mature females and dependent offspring. The captive population of chimpanzees in the United States currently includes over 1300 chimpanzees [9].

Social housing is also recommended for chimpanzees by the Animal Welfare Act and is known to have a positive effect on chimpanzee behavior and health [10,11]. A social partner is perhaps the most important and basic environmental variable for chimpanzees [12,13,14], as it provides constantly changing stimuli and challenges the animal’s social and cognitive functioning. One benefit of social housing is the opportunity for the expression of species-typical social behavior [15]. In a recent survey of chimpanzee experts, the companionship of other chimpanzees was listed as the most important factor for chimpanzee welfare [16].

Social integrations, or the process of introducing unfamiliar chimpanzees to one another to form larger social groups, are often complex in terms of logistics, necessary space, and human-led decisions about which chimpanzees should be introduced to each other. In zoos, research facilities, and sanctuaries, social integrations occur for a variety of reasons, which may include breeding recommendations (zoos), larger enclosures that will accommodate larger groups, the deaths of individuals, or the desire to increase the welfare of the animals. However, social integrations in captivity are also known to be stressful to both the chimpanzees and their caregivers [17]. Cortisol levels, which are indicative of stress, may also be elevated during times of social upheaval, particularly for male chimpanzees [18,19].

Because of the territorial nature and intolerance to strangers of wild chimpanzees, particularly for the males, social integrations in captivity are potentially dangerous. Aggression directed towards new group members is common in chimpanzees, and wounding and injuries sometimes occur [20,21]. Although there are few articles that discuss the deaths of chimpanzees during social integrations, and they are rare, they do occur (first author, personal experience; S. Ross, personal communication). However, most group formations of chimpanzees do have successful outcomes with observed affiliative behavior, which may be one of many possible indicators of a successful introduction [22,23,24,25,26,27,28,29,30,31,32].

Chimp Haven believes that chimpanzees can best thrive in larger, more species-typical mixed sex groups, and houses the chimpanzees in large compatible social groups (current average group size = 11). Chimpanzees at Chimp Haven are housed in groups of at least three and up to 20+ individuals, depending on the housing area and individual compatibility [33,34,35,36]. There are various methods of introducing chimpanzees in captivity to make larger social groups. Introduction methods include: (1) dyadic introductions, which involve testing out a variety of pairings of chimpanzees one at a time before integrating them into a larger group; (2) extended “howdy’s”, where chimpanzees have progressively more access to one another over time, beginning with visual and auditory access and proceeding to protected tactile access; (3) group-to-group introductions, which involve introducing two separate groups of chimpanzees to each other at the same time [37,38,39]. Each method may be successful and which methodology is employed may depend on the facilities available for the integration, the experience level of the staff, and the backgrounds of the chimpanzees. Chimp Haven typically performs group-to-group introductions.

As attrition occurs in existing chimpanzee groups in various captive settings and additional chimpanzees are retired, social integrations will continue to be an important part of captive chimpanzee management. The goal of this study was to provide previously unavailable information on chimpanzee behavior and variables that may affect their behavior during social integrations. This study also aimed to inform captive chimpanzee managers regarding the level of aggressive, affiliative, and submissive social behaviors that might be expected to occur within the first hour of a social integration between unfamiliar chimpanzees.

We made the following predictions for each of the three behaviors examined (aggressive, submissive, and affiliative). For aggressive behavior, the following types of chimpanzees would be more aggressive compared to their counterparts: males, older chimpanzees, or wild born chimpanzees. For submissive behavior, nursery-reared or captive born chimpanzees would be more submissive than their wild born or human-reared counterparts, particularly the males. For affiliative behavior, female, younger, or captive-reared chimpanzees would be more affiliative compared to their counterparts. In addition, we also predicted that the chimpanzees in larger groups including more individuals would exhibit a higher frequency of varied types of behaviors toward others than those in smaller groups. Finally, we predicted that more affiliative behavior would occur in the open-topped areas due to the larger space.

## 2. Materials and Methods

### 2.1. Facilities

Chimp Haven lies on two hundred forested acres outside of Shreveport, Louisiana and consists of a variety of enclosures based on the individual needs of the chimpanzees. Enclosures range from enclosed outdoor spaces with pine straw flooring; enclosed outdoor areas with a natural grass substrate, wire mesh walls, and solid metal brachiating bar ceilings that peak at a height of 7 m; 25-acre open air corrals with cement, wood and mesh walls; to large, multi-acre forested habitats bound by concrete, wood and mesh walls and water moats. Areas that are connected by multiple overhead mesh chutes enable the chimpanzees to be rotated throughout the various areas of the sanctuary. The chimpanzees all have access to adjacent indoor bedrooms, outside of times when the areas are being cleaned or maintained. The chimpanzees also always have access to the outdoor areas, unless there is need for maintenance or during times of inclement weather. All the enclosures have elevated climbing structures and wire shelves and the chimpanzees receive nesting materials of hay, pine straw, blankets or woodwool daily. The chimpanzees are fed a fresh produce diet twice daily. A commercially available primate diet and water is available ad libitum. Enrichment is provided daily, 365 days a year, and the chimpanzees receive additional grain forage on scheduled days.

The facilities at Chimp Haven were designed with the knowledge that many social integrations would need to be conducted in various areas of the sanctuary. Although each introduction is different, the sanctuary co-founders had prior successes introducing chimpanzees in a group-to-group manner [40,41], and most introductions at Chimp Haven fit this model, with some exceptions, particularly when a chimpanzee arrives without groupmates and may be introduced to another individual or pair. Prior to an introduction, multiple aspects of both the individual chimpanzees involved and the environment are considered. Assessments allowed for the optimal placement of the chimpanzees and informed our choices of who to integrate with whom. Facilities, weather, the chimpanzee’s history as well as the physical and psychological states of the chimpanzees and staff on the day of the introduction, are all taken into account. If staff members are not feeling well and are overly anxious, this may impact the chimpanzee’s behavior during a social integration due to the close relationships between the caregivers and chimpanzees. If any of these factors were not optimal, an introduction would be postponed and rescheduled if necessary.

#### Enclosures Utilized during Social Integrations

The chimpanzees were introduced in distinct types of enclosures. Closed locations included enclosure types with a covered top with metal brachiation bars overhead or indoor enclosures, while open ones included outdoors areas with no overhead barriers that were open to the sky (Figure 1 and Figure 2). Open areas were 6000 square feet and closed locations ranged from 395 to 4138 sq. ft. The chimpanzees were not introduced in our multi-acre habitat areas, as a water moat provides a barrier on one end of the enclosure and, given historical accounts of chimpanzees being chased into water moats during aggressive encounters, we felt the risks were too great to conduct social integrations in these areas.

### 2.2. Subjects

The subjects were 282 chimpanzees (male: n = 135; female: n = 147) that were involved in social integration sessions between the years 2005 and 2015. Each social integration involved 2 to 26 chimpanzees (mean = 9) and their age ranged from less than one year old to 59 years old (mean = 30). Some of the chimpanzees were involved in multiple introductions at different ages. The chimpanzees were either born in the wild (n = 56) or born in a captive environment (n = 205), but in some cases we did not have information on where the chimpanzees were born, and these chimpanzees were listed as of unknown birth status (n = 21). The rearing histories of the chimpanzees also varied and included enculturated chimpanzees who were raised in human homes as if they were human infants (n = 4); raised by humans in other settings (n = 7); reared in a research facility nursery (n = 93); mother-reared for more than 6 months, whether in the wild or in captivity (n = 106); or of unknown rearing status (n = 72).

### 2.3. Methods

#### 2.3.1. Data Collection

Introduction data were collected with pen and paper by three trained observers after the doors had been opened between unfamiliar chimpanzees for 225 introductions. The timeframe we targeted for an introduction to last was 60 min. This formal observation timeframe was selected based on our observations that the chimpanzees typically settled down and had reduced numbers of interactions after the initial 60 min of the doors being opened between unfamiliar chimpanzees. However, our observations and data collection sometimes lasted longer than 60 min (mean = 59 min, Range 9–157 min). These durations varied depending upon the individual introduction. Depending on the introduction, a minimum of one to a maximum of three observers collected data. Data were collected ad libitum and observers used portions of Chimp Haven’s standard ethogram, which consisted of two letter abbreviations for various behaviors. Behaviors and their definitions are provided in Appendix A. For this analysis, sexual behavior was included as affiliative behavior. The observers recorded all noticeable behaviors and givers and receivers of the behavior during introductions, using ad libitum sampling. This observation method did not allow the observers to follow a specific individual, record the durations of behaviors, or allow for the notation of all behaviors emitted by a single individual, and so they may have missed occurrences of behaviors. Ad libitum sampling was selected for both practical reasons and for the safety of the chimpanzees involved in the social integrations. Chimp Haven strives to provide the best care possible and always puts the chimpanzees’ health and safety first, especially in potentially volatile situations such as social integrations. Ad libitum sampling allowed us to both monitor the safety of the chimpanzees while recording their behaviors. Given this, it was important to have multiple observers (particularly for larger groups) focusing on the most noticeable behaviors in large introductions when many behaviors were happening at the same time. This was important to catch wounding events, sudden changes in behavior, and note any major events whether affiliative, aggressive, or submissive that might be important, as outlined in the ethogram. Interobserver reliability was not conducted due to the original intent of the sampling, which was chimpanzee safety. The observers in this study were members of Chimp Haven’s behavior department who were extensively trained and tested by the first author on the use of the Chimp Haven specific ethograms and codes, which were utilized for various non-invasive behavioral observation projects at Chimp Haven. At the time of this study, interobserver reliability had been tested, utilizing this same ethogram for another active behavioral observation protocol occurring during the same timeframe. Additional staff members from our veterinary and husbandry departments were also at each introduction in case of any emergencies, and they were also present to monitor the chimpanzees. The date, time, duration and location of the introduction were also recorded.

#### 2.3.2. Data Transcription

All the data were later transcribed onto spreadsheets. As we took data ad libitum, it was possible that multiple observers recorded the same conspicuous behavior (e.g., fighting) involving the same chimpanzees simultaneously. To avoid counting the same behavior involving the same individuals more than once, we inspected the dataset and removed any duplicates. We then categorized observed behaviors into three types of behaviors: aggressive, submissive, and affiliative for analysis.

#### 2.3.3. Data Analysis

To examine whether the subjects’ attributes or environments were associated with the behavior of interest during introductions, we performed generalized linear mixed models (GLMMs) with a negative binomial distribution using the glmmADMB package (Bolker et al., 2012) in R Version 3.4.0, created by R Core Team, Vienna, Austria (2017). Due to the nature of our data collection methods involving ad libitum sampling, as described above, we used the frequency of aggressive, submissive, and affiliative behaviors as our response variable, respectively, instead of rates. The observers were able to capture potentially rare and conspicuous behaviors during social integration but were unable to record all the behaviors emitted by each individual during a set duration of observation time, potentially missing some occurrences of a particular behavior, which made the calculations of rates inappropriate. However, we added the duration of observations conducted during a social integration session or event, and the group size of the introduction event to our models as control variables because they could potentially influence the frequency of each behavior. The variable duration was included in our statistical model, not primarily to analyze it in relation to frequencies as a main variable, but rather since behavioral frequencies tend to increase as durations increase. In some cases, data collection occurred beyond the first hour of the introduction. Given this, it was important to include this variable in the model, although it was not used for the purpose of the main analysis.

Consequently, our model for each prediction included the subject’s sex (male vs. female), age (continuous variable), birth status (captive vs. wild vs. unknown) and rearing histories (enculturated vs. human-reared vs. mother-reared vs. nursery-reared vs. unknown), and the location (closed vs. open areas), group size (continuous variable) and duration (continuous variable) of the introductions as fixed factors. Introduction groups and subjects involved in the introduction were set as random factors.

For each behavior, we fitted the maximal model including all the fixed factors of interest and removed nonsignificant factors using the backward elimination process until all the fixed factors were significant or showed a nonsignificant tendency (*p* < 0.1) to achieve the best model. We also fitted all GLMMs with possible interaction terms (e.g., sex*age, sex*rearing, etc.) and compared them with the models only with main effects, in order to select a better fit model for each behavior based on the lower value of Akaike information criterion (AIC). All models indicated lower AIC values for the models with the main effects only. Consequently, we report the results of the parsimonious models with main effects only as our final models. We assessed the significance of each overall model against the null model based on likelihood ratio tests with the ANOVA function and chi-square distribution. Since we analyzed the same set of data for three predictions, we adjusted our critical levels of statistical significance for overall models using Bonferroni corrections (*p* = 0.05/3 = 0.016).

## 3. Results

A total of 225 introductions involving 282 individuals (females: n = 147; males: n = 135; age range: less than one year old–59 years old) were analyzed. Of the 225 introductions, all were considered successful introductions. At the time of this study, Chimp Haven defined successful introductions as an individual remaining in group for at least 6 months, with only minor wounding (after initial introduction event) and no chimpanzee being removed from the group for any reason, including behavioral evidence of distress or inaccessibility to food or water, or indoor space, and each individual interacting and playing an active role in their social group based on ongoing monitoring and behavioral observations by animal care staff. The definition of a successful introduction is very important, and there is a distinction to be made between an introduction session or event and long-term introduction success. Major wounding during the initial event would require the separation of the group for the treatment of any injuries, which would then indicate that an introduction was unsuccessful.

During this timeframe, from 2005 to 2015, there were three unsuccessful introductions that were not included in the analysis, as observations had to be suspended so that staff could focus on separating the groups for the safety of the chimpanzees. In each case, the chimpanzees involved went on to be successfully introduced and live in other groups at a later time. Due to this success rate and a lack of complete observational data, we were unable to make comparisons between successful and unsuccessful integrations. Other behaviors we were unable to analyze due to low frequencies of occurrence were abnormal behavior and play. Each introduction included 2 to 26 chimpanzees and lasted for the duration of 59 min on average. After an initial successful introduction session, the chimpanzees that had been introduced remained together and were not separated; however, more chimpanzees may have been added in an additional session. Although formal observations ended after approximately an hour for each integration, staff continued to closely monitor the new chimpanzee groups until all chimpanzees were observed to access food, water, and indoor areas, as well as continuing to monitor wounding and group interactions in the following months. Eighty-three introductions were conducted in “open” locations, while 142 introductions occurred in “closed” locations. Overall, affiliative behaviors were observed more frequently than either aggressive or submissive behaviors. Males showed behaviors in all three behavioral categories more frequently than females (Figure 3).

### 3.1. Aggressive Behavior

All the factors except for group size were associated with the frequency of aggressive behavior observed during introductions (χ^2^(10) = 59.42, *p* ≤ 0.0005) (Table 1). Males were more likely to be aggressive compared to females. Older chimpanzees were more aggressive than their younger counterparts. Captive born and human-reared chimpanzees tended to be more aggressive than wild born and nursery-reared chimpanzees, respectively. There were more aggressive behaviors when the social integrations occurred in open locations vs. closed ones. Aggression was not associated with the group size of the introductions. Females directed aggression equally toward other females and males, while males directed aggression toward other males much more than toward females (Figure 4). Display, chase, hit, hit at, and other aggressive behaviors were the most observed aggressive behaviors (Figure 5a).

### 3.2. Submissive Behavior

The subject’s sex and rearing history, and the group size and duration of introductions were associated with the frequency of submissive behaviors (χ^2^(7) = 61.96, *p* ≤ 0.0005) (Table 1). Males exhibited more submissive behaviors than females (Figure 3). Nursery-reared chimpanzees tended to show more submissive behaviors than their mother-reared counterparts. Chimpanzees exhibited more submissive behaviors when the group size was smaller. Unlike the other two behaviors, submissive behaviors were not associated with the location of the introductions. The most frequently observed submissive behaviors were avoid, fear grimace, pant grunt, bob, and present (Figure 5b).

### 3.3. Affiliative Behavior

The subject’s sex, and the location, group size and duration of introductions were associated with the frequency of affiliative behaviors observed during introductions (χ^2^(4) = 56.62, *p* ≤ 0.0005) (Table 1). Males exhibited more affiliative behaviors than females (Figure 3). The chimpanzees exhibited more affiliative behaviors in open locations compared to closed ones, as well as when the group size of the introductions was smaller. Approach was the most frequently observed affiliative behavior, followed by other affiliative behaviors such as mouth, follow, and touch (Figure 5c).

### 3.4. Location, Group Size, and Age Differences

Affiliative and aggressive behaviors were observed more frequently in open locations vs. closed ones. Submissive behavior was not significant in relation to the location (Figure 6). When a smaller number of individuals were introduced together, affiliative, and submissive behaviors were observed more frequently than in larger groups of more than twelve individuals (Figure 7). Aggressive behaviors were not impacted by group size. Older individuals were more aggressive during introductions than their younger counterparts.

## 4. Discussion

Overall, social integrations at Chimp Haven have been overwhelmingly successful and hundreds of chimpanzees have been integrated into new or larger families. With captive chimpanzees moving to new locations for breeding, as various facilities close, as well as with the continued movement to sanctuaries [42,43,44,45], social integration will continue to be an important part of providing and promoting positive welfare for the species. In addition, as the captive chimpanzee population ages and faces attrition, particularly in research facilities and sanctuaries where breeding no longer occurs, social integrations will continue to remain important tools for socializing chimpanzees whose groups may become smaller due to the deaths of group members.

With so much of the literature on social integrations in captivity being focused on stress, wounding, and aggression [36,46,47], as well as the knowledge of chimpanzee aggression towards strangers in the wild [48,49] and the propensity for the males to inflict potentially fatal wounds [50,51,52], it was somewhat surprising that affiliative was the most frequently observed behavioral category during the first hour of social integrations in this study in both male and female chimpanzees. However, female chimpanzees do emigrate to new groups and must find ways to successfully do so in the wild [53,54], and males frequently interact with one another in positive ways, although they remain competitive [55,56]. Although we do not have complete introduction histories for all of the chimpanzees at Chimp Haven, it is also likely that many of this captive population has undergone social integrations in their past and may have some experience with changing social groups either due to births, deaths, or even research protocols that required them to be removed from and then reintegrated into social groups [57,58,59]. In another study, when a female chimpanzee was integrated into an existing group at Chester Zoo, they found that each chimpanzee was involved more often in affiliative interactions per hour (mean = 0.79) than agonistic interactions (mean = 0.50) [60]. The chimpanzees in this study were very interested in one another, with the most common affiliative behavior being approach and contact behaviors such as mouth and touch occurring multiple times during the first hour of being introduced. These behaviors may also have occurred after aggressive interactions as a form of post-conflict affiliation [61]. Other frequently observed behaviors in each category were normal and appropriate behaviors for chimpanzees in social situations. Displaying, chasing, and non-contact aggression are typical for the species during agonistic encounters, and avoidance, fear grimacing, pant grunting, and bobbing are also species-typical submissive behaviors.

### 4.1. Sex Differences

Numerous studies have examined sex differences between male and female chimpanzees; many of these relate to tool use and hunting, but others have looked specifically at social behavior [62,63]. The males in this study engaged in all three types of behavior (aggressive, submissive, and affiliative) more often than the female chimpanzees. In a study that looked at male and female infants at Gombe, the males interacted with significantly more individuals than the female infants [64], and wild adult male chimpanzees are more gregarious than females [65].

#### 4.1.1. Aggressive Behavior

Males were more aggressive than females in the first hour after being introduced to new chimpanzees. This result is not surprising given the volatile nature of male chimpanzee relationships in both the wild and captivity as they navigate the dominance hierarchy [66,67]. In both the wild and captivity, male chimpanzees are involved in dominance displays and disputes more frequently than females. Chimpanzee males with their loud boisterous displays and propensity to patrol the boundaries of their territories have long been thought of as the more aggressive of the two sexes and, indeed, in both captivity and the wild, there have been incidents where males have killed other males [50,51,52]. More commonly, males have been known to injure other males in dominance struggles [47]. In another study at Chimp Haven that looked specifically at wounding rates during social introductions, males received 64% of all wounds, and larger groups with more males experienced significantly more wounding [36]. In another study of social integration, males also directed more agonistic behaviors at others than females; however, they were not specifically directed towards other males, as we observed in this study [60].

#### 4.1.2. Submissive Behavior

Female chimpanzees are known to be submissive to all male chimpanzees in the wild [68]; however, in captivity, this is not always the case, and, in some cases, females may even be the dominant members of their groups [69]. In this study, the males displayed more submissive behavior than the female chimpanzees, perhaps due to their desire to form alliances and coalitions with new group members. We also found that nursery-reared chimpanzees tended to be more submissive than those who were mother reared. This was expected, given previous findings that nursery-reared chimpanzees display more submissive and less prosocial behavior in social settings and that early deprivation may have lasting effects on chimpanzee social behavior [23,24,27,40,70,71]; but in contrast, see [72].

#### 4.1.3. Affiliative Behavior

Males also showed higher frequencies of affiliative behavior than females in the first hour after being introduced to unfamiliar chimpanzees. Males in the wild and captivity, although competitive, often show high levels of cooperation and grooming [73,74], which likely helps to forge strong bonds and strengthen and reinforce any alliances.

### 4.2. Rearing and Birth Status Differences

Captive born and human-reared chimpanzees tended to display more aggressive behavior during social integrations. Chimpanzees that are wild born may have more social skills due to being exposed to larger social groups at younger ages and may, therefore, engage in lower rates of aggression, while captive born chimpanzees may have lived in smaller social groups. Human-reared chimpanzees may have had limited contact with conspecifics in their early years and, therefore, have limited social experience. For submissive and affiliative behaviors, there were no differences between captive and wild born chimpanzees. In a recent study that looked at differences in sociability between captive and wild born chimpanzees, there were no differences in grooming behavior between the two groups or based on rearing history [75]. The authors postulate that living in social groups may help to ameliorate the consequences of human rearing.

### 4.3. Location (Closed vs. Open), Group Size, and Age Differences

Introductions in the open topped areas were associated with higher frequencies of both aggressive and affiliative behavior than those in the closed areas. Open topped areas were larger than closed topped areas and required more observers; this may have led to observations of more behaviors in these areas. However, an alternative explanation may be that the additional space in the open topped areas provided the chimpanzees with more opportunities to engage in both aggressive conflict and then the resolution of that conflict through affiliative behaviors. Closed topped areas provided additional escape routes via brachiation bars and may have allowed the chimpanzees to avoid interacting with one another as frequently. In contrast to this study, chimpanzees that were moved between larger open topped corrals to closed topped Primadomes with less square footage did not exhibit any significant changes in aggressive, submissive, or affiliative behavior [76]. Self-directed behaviors were reduced in another study when the chimpanzees had more accessible areas during introductions but did not change with an overall increase in space [77], suggesting that having multiple areas available during social integrations may be more important.

Smaller groups of less than 12 individuals experienced more submissive and more affiliative behaviors during the first hour after being integrated. Surprisingly, group size did not impact aggressive behavior. Given that there are many more chimpanzees to meet in a larger group, it is possible that it took more time to develop aggressive interactions. In some studies that looked at social integrations over time, they witnessed an increase in agonistic behavior over time [60,75,78]. In this study, groups of seven to 12 individuals were observed to engage in the highest levels of affiliative behavior lending additional support to the recommendations of both the AZA and the NIH that chimpanzees should be maintained in groups of seven to nine chimpanzees. Prior studies have predominantly examined individual or small groups of chimpanzees that have been introduced to one another, making it difficult to determine the effects that various group sizes might have on behavior. However, group sizes of seven or more chimpanzees have influenced affiliative behavior, locomotion, and personality traits in two separate studies. In one of these studies, these larger group sizes resulted in higher ratings on sociable, curious, and playful personality traits [79]. In the other study, chimpanzees in groups of seven or more with at least half males showed increases in affiliative behavior, but not aggression [13].

Older individuals were more aggressive during introductions than their younger counterparts. Subadult males between the ages of 10 and 19 years were more aggressive than other age/sex categories. However, in one study, adolescent males received more wounds but did not cause more wounds than other group members [80]. Older male chimpanzees may need to incorporate themselves within the hierarchy quickly, necessitating their need to learn as much social information as possible through aggressive interactions in as short a period as possible. Juvenile males and females, who may still be lower in rank, may be more interested in forming alliances and friendships. In another study, juveniles seemed to play a role in encouraging contact between the members of two groups that were introduced [81]. Older females may be less likely to welcome newcomers to their groups; however, younger females may be easier to integrate due to their need to emigrate and become members of new groups in the wild. In other studies that looked at behavioral differences due to age, chimpanzees over 30 years old showed less aggression overall than younger chimpanzees; however, this was not in the context of social integrations [82], while in another study, chimpanzees over 35 were less affiliative than younger chimpanzees [83].

The authors recognize that this study does not assist in predicting the outcome of an introduction, which would be very beneficial to the management of captive chimpanzees. Our goal with this particular study was to understand the behaviors to be expected in an introduction between previously unfamiliar chimpanzees and not to predict the final outcome (success/failure) of a particular introduction. However, given the limited amount of research available on chimpanzee introductions, we feel that this study provides some practical and useful information for the management of captive chimpanzee social integrations.

This study demonstrates that while aggressive encounters are to be expected during chimpanzee social integrations, we can also expect substantial affiliative behaviors even with this highly territorial species. Due to the potentially volatile nature of chimpanzees and the expectation of aggression, managers may be overly cautious of natural aggressive behaviors during introductions. This, as well as differences in methods regarding the speed and style of forming chimpanzee groups, may lead to hesitation to form larger multimale/multifemale groups, which may be optimal for overall chimpanzee welfare. Of course, there are exceptions, and smaller groups are sometimes best for individual chimpanzees with special needs.

Our data demonstrate that a range of behaviors are expected, and observing these behaviors is normal for an introduction between unfamiliar chimpanzees. The study further illustrates that an hour is an adequate amount of time to see a wide range of social behaviors displayed by chimpanzees during an introduction session. In addition, aggressive encounters, including fights, are likely and should not necessarily indicate the need to end an introduction session or separate the individuals involved. The absence of these behaviors during an introduction session may be indicative of an issue that may or may not be able to be resolved, but further research would be required to elucidate this.

Further, while not outlined in this study, the success or failure of an introduction may require more nuance in the absence of major wounding events. The success of an introduction cannot always be determined in the first hour after the chimpanzees have been introduced; a group remaining together for 6 months may not be long enough for larger groups. Some reasons for this may include changes in a chimpanzee social group over time due to illness, the death (or birth) of a groupmate, or hierarchical changes that may impact social relationships within the group. The results of this study may also be useful for understanding the importance of surveying individual (age and sex) and group (group size) attributes before any type of social introductions. For example, if males are being introduced to males, more aggression might be expected, and more precautions may be warranted. Due to the limited research on this particularly important captive chimpanzee management topic, the authors felt that various types of facilities might gain valuable information on expected behaviors and use that to their advantage in chimpanzee introductions where other factors will likely vary. It is our hope that this study may also serve as a starting point for others to compare to during both future successful introductions, as well as during unsuccessful introductions. For example, if there are low rates of these behaviors in an initial introduction, or if they are non-existent, could that lead to a potential predictor of a future unsuccessful outcome for an introduction?

## 5. Conclusions

The current study provides evidence that sanctuary chimpanzees express aggressive, submissive, and affiliative behaviors in various ways according to their sex, age, rearing history, and the group size in which they reside during social integration.

Males were more socially interactive during introductions than females, which included being more aggressive than females generally. However, the most surprising result was the high levels of affiliative behaviors during introductions. While in some ways the males reacted as predicted, there were many other factors (group size, type of enclosure, and age) that impacted the outcome of an introduction within the first hour of full contact. Groups of 7–12 individuals displayed the most affiliative and submissive behaviors during introductions, while large open topped enclosures increased the frequency of both aggressive and affiliative behavior. Older chimpanzees were more aggressive than their younger counterparts but were still able to be successfully introduced.

This study suggests that captive chimpanzees may behaviorally cope with a stressful life event such as social integration based on their social attributes or environmental settings. Such knowledge may aid in the captive management of chimpanzee introductions. When planning social integrations of two or more unfamiliar chimpanzee groups, this study provides captive managers with factors that may contribute to successful outcomes and may minimize the catastrophic results of social integrations such as severe injuries or deaths.

Further research with a more systematic data collection method beyond ad libitum sampling, comparing similar behaviors with other captive/sanctuary settings, and considering other important variables that could not be included in this study, such as individual rank, the temporal patterning of behaviors, and the occurrence and severity of wounding and aggression, would provide additional insight into the factors contributing to general, successful, or unsuccessful chimpanzee social integrations. Finally, this study was based on an overwhelmingly high number of successful social integration events. Future studies including or focused on unsuccessful social integrations would provide comparative data to the current study.

## Figures and Tables

**Figure 1 animals-12-02421-f001:**
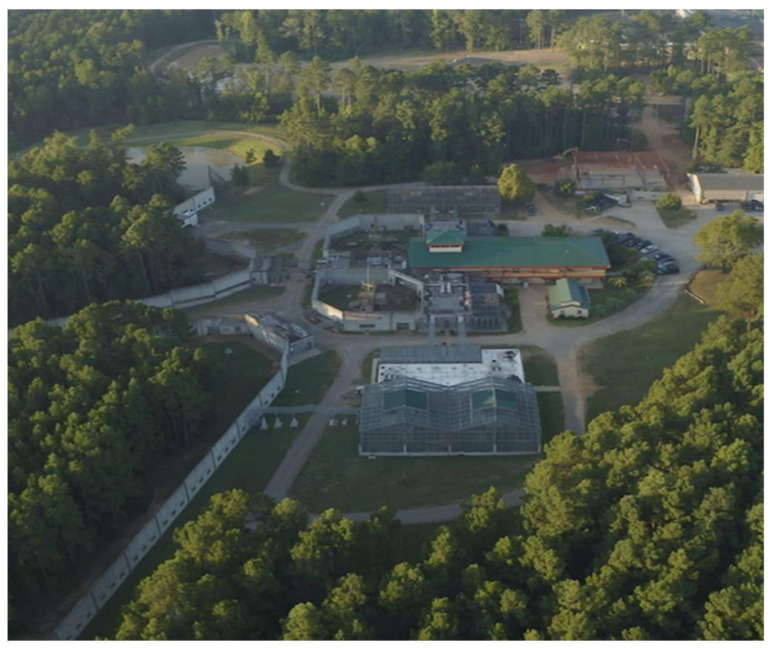
An aerial drone photo of Chimp Haven showing two open topped corrals in the middle of the photo and closed topped enclosures with brachiation bars at the top and bottom of the photo. Sections of Chimp Haven’s three multi-acre forested habitats are shown on the left side of the photo.

**Figure 2 animals-12-02421-f002:**
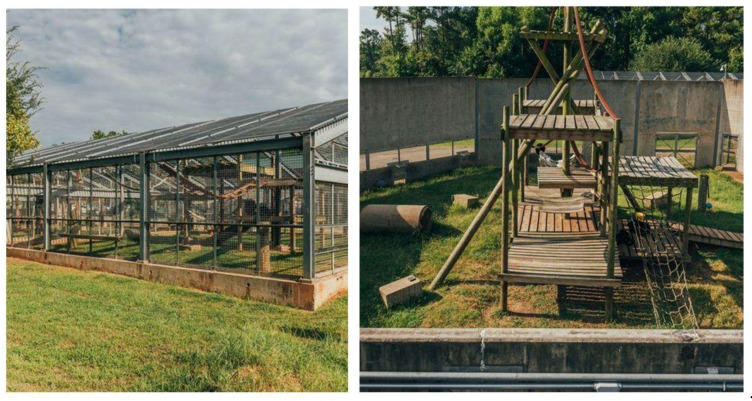
L**eft**: A photo of a closed topped enclosure with brachiating bars at Chimp Haven used for social integrations. **Right**: A photo of a large open topped corral at Chimp Haven utilized for social integrations.

**Figure 3 animals-12-02421-f003:**
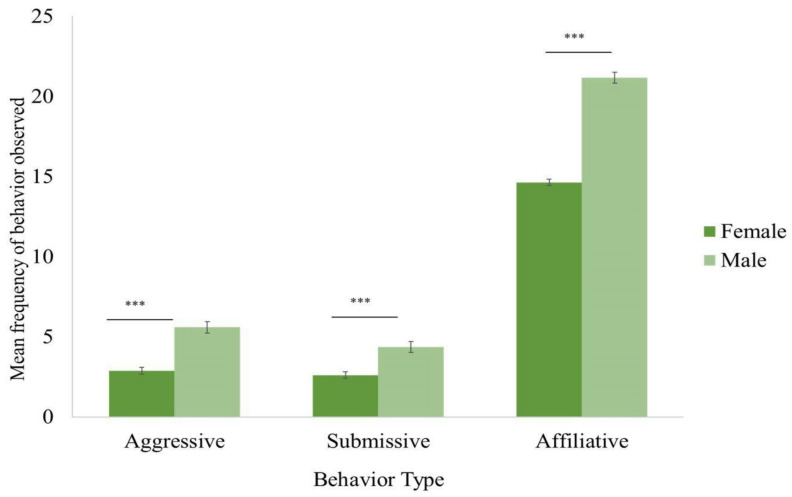
Mean frequency of aggressive, submissive, and affiliative behaviors observed during social integration by males and females. In all three behavior types, males exhibited more behavior than females. The critical level of the statistical significance was set at *p* ≤ 0.016 after a Bonferroni correction: *** Significant at *p* ≤ 0.0005.

**Figure 4 animals-12-02421-f004:**
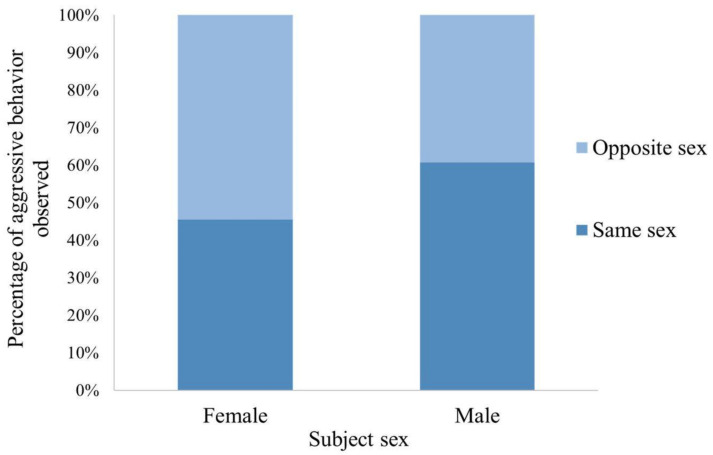
Percentage of aggressive behavior directed toward the same sex vs. opposite sex during social integration by the subject’s sex. More than half of male aggression was directed toward the same sex conspecifics (other males), while females directed aggression towards the opposite sex (males) more than the same sex (females).

**Figure 5 animals-12-02421-f005:**
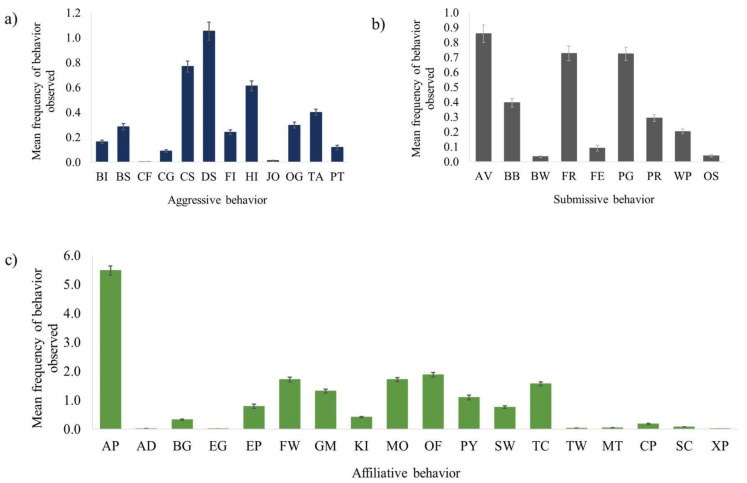
(**a**–**c**). The breakdown of aggressive, submissive, and affiliative behavior types observed during the first hour of social integration by all chimpanzees. Each of the figures (**a**–**c**) show the mean frequency of behavior observed overall per behavior type: (**c**) affiliative behavior; (**b**) submissive behavior; (**a**) aggressive behavior.

**Figure 6 animals-12-02421-f006:**
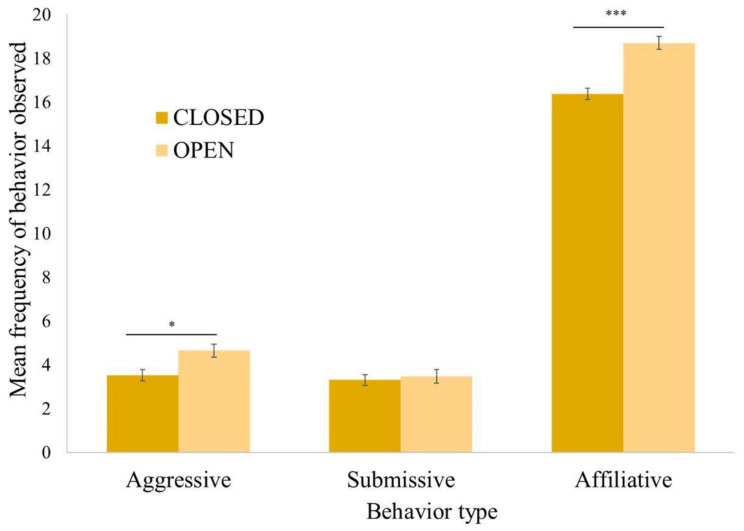
Mean frequency of aggressive, submissive, and affiliative behavior types observed overall during social integration by closed vs. open top locations. The critical level of the statistical significance was set at *p* ≤ 0.016 after a Bonferroni correction: * *p* ≤ 0.016, *** *p* ≤ 0.0005. Aggressive and affiliative behaviors were more frequently observed during social integration in open top locations than in closed locations. This was particularly the case for affiliative behavior.

**Figure 7 animals-12-02421-f007:**
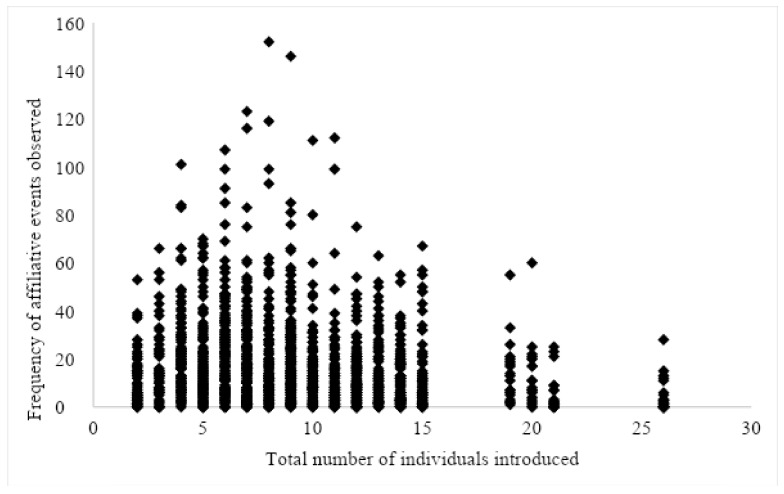
Scatterplot with frequency of affiliative events related to group size or total number of individuals introduced during a social integration.

**Table 1 animals-12-02421-t001:** Parameter estimates from the GLMMs that produced significant results.

Behavior	Significant Factor	Factor Level ^a^	Estimate	S.E.	Z	*p* Value
Aggressive	Sex	Female	Reference			
		**Male *****	0.623	0.099	6.28	**0.0000**
	**Age**	**(Continuous) ****	0.017	0.006	2.69	**0.0073**
	Birth status	Captive born	Reference			
		**Wild born ^(^*^)^**	−0.404	0.212	−1.90	**0.0570**
		Unknown	−0.2087	0.216	−0.97	0.3300
	Rearing history	Nursery	Reference			
		Enculturated	0.521	0.382	1.36	0.1723
		**Human-reared ^(^*^)^**	0.530	0.274	1.93	**0.0530**
		Mother-reared	0.183	0.141	1.30	0.1947
		Unknown	−0.086	0.139	−0.62	0.5343
	Location	Close	Reference			
		**Open ***	0.3207	0.127	2.53	**0.0113**
	**Duration**	**(Continuous) ***	0.674	0.288	2.34	0.0192
Submissive	Sex	Female	Reference			
		**Male *****	0.378	0.080	4.76	**0.0000**
	Rearing history	Nursery	Reference			
		Enculturated	−0.086	0.322	−0.27	0.7889
		Human-reared	−0.254	0.225	−1.13	0.2593
		**Mother-reared ^(^*^)^**	−0.154	0.093	−1.66	**0.0978**
		Unknown	−0.160	0.104	−1.53	0.1259
	**Group size**	**(Continuous) *****	−0.050	0.010	−5.03	**0.0000**
	**Duration**	**(Continuous) *****	0.735	0.185	3.98	**0.0001**
Affiliative	Sex	Female	Reference			
		**Male *****	0.264	0.071	3.74	**0.0002**
	Location	Close	Reference			
		**Open *****	0.30598	0.081	3.79	**0.0002**
	**Group size**	**(Continuous) *****	−0.05388	0.009	−5.88	**0.0000**
	**Duration**	**(Continuous) *****	0.457	0.160	2.85	**0.0043**

Only significant factors for each prediction are included in this table. ^a^, *****
*p* ≤ 0.016 (after a Bonferroni correction); ******
*p*≤ 0.01; *******
*p*≤ 0.0005; **^(^*^)^** *p* ≤ 0.1 (non-significant tendency).

## Data Availability

The data presented in this study are available on request from the corresponding author. The data are not publicly available due to restrictions regarding data sharing.

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
