# Peer review of "Aggressive, Submissive, and Affiliative Behavior in Sanctuary Chimpanzees (Pan Troglodytes) During Social Integration"

_animals, 2022, doi:10.3390/ani12182421_

Round 1

Reviewer 1 Report

The authors of the study “Aggressive, submissive, and affiliative behavior in sanctuary chimpanzees (Pan troglodytes) during social integration” did a fantastic job organizing the data collection of such a massive project over such a long time frame. Studies like this one, collecting long term data are extremely valuable and needed, even more so taking into account the massive number of chimpanzees included. Furthermore, they also managed to present their data and explain the importance and difficulty of the study topic in a very precise and interesting manner.

I highly recommend to publish this article in your journal and simply add some comments to help make it more reader friendly and some suggestions to potentially acquire even more insights based on this data set (although not necessarily within this article).

11 “Chimpanzees moving into sanctuary settings often have lived in smaller groups at their previous facilities.” – I understand what the authors wish to say here, but I suggest to rephrase this part slightly as this would not be the typical trend for all sanctuary intakes, but rather for those coming from research facilities (most of the chimpanzees from this massive study population). Chimpanzees directly confiscated from poaching in Africa or those coming from the pet and entertainment industry do not tend to live in small groups necessarily.

29 The authors always refer to “Ad libitum” observations, which generally might be the case having recorded more than only social interactions. However, at least the data recording of social interactions (the only data used in this study) seems well structured and based on a well-defined ethogram. If this is the case, I would refer to the methodology used in this study as multifocal “All occurrence” focused on social behaviors, rather than ab libidum.

However, I find that the naming of certain methodologies often varies greatly between different countries, as such this is merely a suggestion.

42 “ In captivity” – here again, I think the author actually mean to refer to captivity in research institutions?!

162: close to the section about the Subjects I would welcome a more precise explanation of the Integration session or rather what happens afterwards (as observations are limited to 60min but the integration session actually continues). Although the authors describe several options of integration procedures (very much appreciated) it might not be clear for all readers how integration sessions are structured in this sanctuary or rather what happens after the 60min observations. Some doubts I had after reading these sections: After a successful session (after the observation ended) individuals would stay together permanently and would not be separated anymore or are there repeated integration sessions? However more individuals could be added in later sessions? Or the individual could be moved (after 6 month) to another group?

Chimpanzees would be permanently housed together after successful session: A successful session being the first encounter not turning into an excessive aggressive conflict? Is there any information on night wounding (although being mild wounding) or general social behavior frequencies during the next days?

I believe adding a few lines of information here would make the results even more clear and especially helpful to readers interested in taking advantage of these insights for practical use in other institutions. However I am not suggesting to change any categories, statistics or results here, but simply suggest to add more information for a better understanding.

202 I am surprised to see that the predictor values were not converted to rates. This would be more comfortable in order to compare the data with other studies, but is not a necessity for this study itself as I understand that the observations all approximately lasted for an hour (mean = 59 min).

As such I am wondering if the duration value analyzed in the GLMM actually refers to the duration of the observation or the duration of the integration session in total and in which case it would be interesting to understand what marks the end of a integration session (not observation). Please add information to clarify within the draft

262 Table 2. and 269 Figure 5: It would be much more comfortable for the reader to split this table a figure and add them separately to their corresponding paragraph. Just to make it more reader friendly.

262 Table 2.: All three predictors are listed to be significantly impacted by the duration of the integration sessions. This is also further down mentioned in the result description; however, I do not find any more information regarding how it is affecting (example: the longer the sessions the more social interactions were scored?) nor any argumentation/discussion to why this might be the case.

The discussion and conclusion is very well structured and well discussed. Once again I want to congratulate the authors for being able to conduct such an important and complicated project.

In continuation I wish to add some thoughts that might be interesting to look into in the future (not for this draft). Possibly this might be possible even with this data set itself:

It might be interesting to look into “missing” dyadic interactions, i.e. if and how many chimpanzees would not engage at all with each other during these initial 60min. Some chimpanzees might avoid each other for a rather long time (far more than 60min actually), especially in sessions of bigger group sizes.

I completely understand that the authors could not make a comparison between successful and unsuccessful integration as for the reasons mentioned (amazing record of successful integrations showing how well-trained staff members are and how well the facilities were designed). However, I think it would be extremely interesting to make a comparison of integration sessions without any wounding at all or even without any aggressive interactions, in comparison to those sessions that included some aggressive interactions and/or wounding.

Reviewer 2 Report

This is an interesting paper that explores a very important topic; social integration of chimpanzees into complex social groups.  The authors collected ad libitum behavioral data during the first 60 minutes of over 200 social integrations.  These social integrations were conducted in a “group-to-group” manner and involved introductions ranging in size from 2 to 26 animals.  The overwhelming majority of these social integrations were considered successful.  The authors identified a variety of differences in aggressive, affiliative, and submissive behaviors during the first 60 minutes.  Sex, age, and integration enclosure type were some of the factors that were related to the observed differences.  While many of the behavioral differences achieved statistical significance, they may not have had much biological meaning.  This statement is based on the fact that almost all of the introductions were deemed successful.  Therefore, the identified differences were not at all predictive of the success or lack thereof, of an introduction.  Because of this, the data are unlikely to be particularly useful.  Someone trying to introduce chimpanzees to one another at another facility will not be able to use any of the findings from this paper to help them predict, after observing behavior for the first hour, whether a social integration will ultimately be successful.  This reduces the value of this manuscript to just a straightforward description of chimpanzee behavior during the first 60 minutes of an introduction.

Some other general comments/critiques:

The authors are quite familiar with the nuances of chimpanzee social introductions.  Sixty minutes of observation per introduction seems like too short a period for the collection of meaningful data.

Again, the authors are familiar with the difficulties of accurately observing the behaviors of large chimpanzee groups.  Are they confident that all of the behaviors of interest could be observed, especially when 10 or more chimpanzees were being introduced?  Were analyses of interobserver reliability performed?

Additional specific comments:

Abstract

Line 27                 134 males plus 147 females equal 281, not 282

Line 32                 What does “associated” mean?  Significantly correlated?

Line 34                 What does “associated” mean?  What does “varied’ mean?

Lines 35-36         The inability to compare successful to unsuccessful integrations was also due to the fact that data collection was suspended during the three unsuccessful integrations.

Introduction

Lines 42-43         This statement is pretty vague, no citations are provided, and it may not be accurate, depending on the way that “typically” is defined.

Lines 46-61         It might make sense to distinguish between AZA and NIH recommendations a little more clearly in this paragraph.  Obviously, with NIH not allowing breeding, groups with dependent offspring are not possible.

Lines 77-79         This sentence really does not fit here.

Lines 85-87         This sentence implies that a successful group formation results in observed affiliative behavior.  This seems like it would be one small indicator of successful group formation, not the defining criterion.

Lines 88-89         Why are only data from 2005-2015 included, if there are another 75 or so introductions involving another 100+ animals?  Are there data available for these more recent social integrations.

Lines 106-107     It is not entirely clear how the current study provides information useful for future social integrations.  Any variations in behavior due to the factors studied in this paper were pretty much inconsequential, as most (almost all) social integrations were successful.

Lines 108-118     While all of these predictions appear sensible, their implications for social integration are not as clear.  The value of observing and assessing these behaviors for those performing social integrations in the future is not apparent.

Materials and methods

Lines 139-140     It would appear that the “group-to-group” introduction technique was used for most/all of the social integrations.  Is this correct?

Obviously, all available information was assessed prior to beginning a social integration.  Were any animals not integrated based on this type of assessment?  Based on their very high success rate, the authors were obviously able to use the information they had quite beneficially.

Line 163               134 males plus 147 females equal 281, not 282

Lines 176-184     Were the data collected on a computer using software or were the data collected by pen and paper techniques?  28 affiliative behaviors, 20 aggressive behaviors, and 12 submissive behaviors seem like a lot to keep track of, even if many of these behaviors rarely, if ever, occurred during the first 60 minutes of a social integration.  Durations of behaviors were not recorded, correct?  How was Interobserver Reliability assessed?

Results

Line 229               The definition of a successful introduction is pretty important.  Does this line imply that major wounding during the initial introduction event would still be categorized as a successful introduction?

Line 233               Are data available for introductions between 2015 and 2021?  If so, these data should be included.

Line 238               While it is obvious that the safety of the animals should come first and the title of the paper does not focus on successful vs. unsuccessful introductions, the real value of this paper should be related to the behavioral indicators in the first 60 minutes of an introduction that signal a successful vs. an unsuccessful introduction.  Unfortunately, if the aggressive, submissive, and affiliative behaviors observed in the first 60 minutes of an introduction are not predictive of anything, then they are not particularly meaningful, diminishing the applied value of this manuscript.

Are the values represented in Figures 3, 5, and 6 the means per individual male and female or means per introduction across all males and females involved in the social integration?

Discussion

Sounds ok.

Conclusions

Lines 454-459     It is unclear how this study informs social integrations of captive chimpanzees.  All 225 social integrations where data were collected were successful, suggesting that all of the identified differences (sex, age, group size, etc.) were of no consequence to the success of social integrations.  Therefore, the value of the data collected and analyzed for this study appears limited.

Lines 466-468     Do the authors have data on unsuccessful introductions?  Such data would indeed be valuable.  But if observations are suspended when social integrations do not go well, how can these data be collected?

Reviewer 3 Report

The authors present an interesting paper where they analyse 10 years of data on chimps introductions in a sanctuary, evaluating different factors that can affect the success. The manuscript is generally well written, and the methods are correct. The authors, however, should improve key parts of the manuscript to reach a larger audience.

1. Simple summary and abstract should be improved. Missing a broad introduction and a simple results section in the simple summary. Can your findings be extended to translocations of wild animals and introductions in other sanctuaries or just introduction of chimps in your sanctuary? The abstract is unbalanced, too much space dedicated to the aim and methods, not enough to broad introduction, results, and discussion (that is the most important part for the broad readership)

2. You would need a broader introduction section before going into chimps saying how animals in sanctuaries can "simulate" social structures present in the wild (and the implications for animal welfare and potential reintroductions in the wild). The introduction is just focused on chimps and that will reduce the readership. 

3. You make predictions but you do not support them with literature. Should add why you made those predictions

4. Table 1 can go in the appendix as it is large and stops the narrative

5. Lines 295-296. Instead of Fig 7 with frequencies you should do a statistical test if you want to support this statement. Also the following statement is not supported by stats.

6. In the discussion/conclusion you should expand on some broader applications. Can your findings help chimp translocations in the wild? 

Round 2

Reviewer 2 Report

Again, this is an interesting paper that explores a very important topic; social integration of chimpanzees into complex social groups. The authors have addressed most of the comments on the original version of the manuscript.  The reasons for not including the social integrations performed after 2015 are not particularly convincing, so the authors should probably eliminate any mention of these additional introductions (Lines 92-93).  Questions still remain concerning the accuracy of behavioral observations when 10+ animals are introduced to one another and there are 1-3 observers recording multiple dimensions of behavior using pen and paper techniques.  This is not an easy task and the authors mention several times that multiple behaviors were likely to have been missed.  Overall, many aspects of the Discussion section seem somewhat speculative and any conclusions related to the relationship between frequencies of aggressive/affiliative/submissive behaviors during the first 60 minutes of a social integration and the ultimate success of social integrations are especially so.

Additional, somewhat nitpicky comments:

Introduction

Lines 51-52         Fission-fusion social organization is not “unique” to chimpanzees.

Lines 92-93         It might be better not to mention the additional introductions (approximately 75 more), if data from those introductions cannot be included in this manuscript.  Alternatively, it might be relevant to present the number of social integrations that were not successful during that period (3 from the first 228 between 2005-2015 and X for the next 72+ social integrations after 2015).

Methods

Lines 154-156     It is unclear why the physical and psychological states of the staff on the day of introduction would be taken into account.

Lines 219-221     Since this study took place over a 10-year period, did the other active behavioral observation protocol (used for interobserver reliability) also occur over a 10-year period?

Discussion

Lines 505-508     It seems that hesitation is unlikely to be related to the formation of large groups; the hesitation is likely to be related to the speed and style used to facilitate the social integrations.

Lines 511-512     Do the authors have any data on the temporal patterning of the three types of behavior?  For instance, does most of the affiliative/aggressive/submissive behavior occur in the first 5 minutes?  10 minutes?  Etc.

Conclusions

Line 537               “Express” may be a better word than “use” in this sentence.

Reviewer 3 Report

I do not have further comments. I believe the authors missed an opportunity to expand on the broader applicability of their data, as also highlighted by another reviewer. But I also need to consider that the paper fits the SI so it is up to the editors of the SI if they want more links to translocations in the wild.
